# ON LINEAR REPRESENTATIONS AND PRETRAINING DATA FREQUENCY IN LANGUAGE MODELS

**Jack Merullo**[◇]    **Noah A. Smith**[♡♣]    **Sarah Wiegreffe**[*♡♣]    **Yanai Elazar**[*♡♣]

[◇]Brown University, [♡]Allen Institute for AI (Ai2), [♣]University of Washington
[*]Co-senior authors.
jack_merullo@brown.edu, {noah, sarahw, yanaie}@allenai.org

## ABSTRACT

Pretraining data has a direct impact on the behaviors and quality of language models (LMs), but we only understand the most basic principles of this relationship. While most work focuses on pretraining data's effect on downstream task behavior, we investigate its relationship to LM representations. Previous work has discovered that, in language models, some concepts are encoded 'linearly' in the representations, but what factors cause these representations to form (or not)? We study the connection between pretraining data frequency and models' linear representations of factual relations (e.g., mapping France to Paris in a capital prediction task). We find evidence that the formation of linear representations is strongly connected to pretraining term frequencies; specifically for subject-relation-object fact triplets, both subject-object co-occurrence frequency and in-context learning accuracy for the relation are highly correlated with linear representations. This is the case across all phases of pretraining, i.e., it is not affected by the model's underlying capability. In OLMo-7B and GPT-J (6B), we discover that a linear representation consistently (but not exclusively) forms when the subjects and objects within a relation co-occur at least 1k and 2k times, respectively, regardless of when these occurrences happen during pretraining (and around 4k times for OLMo-1B). Finally, we train a regression model on measurements of linear representation quality in fully-trained LMs that can predict how often a term was seen in pretraining. Our model achieves low error even on inputs from a different model with a different pretraining dataset, providing a new method for estimating properties of the otherwise-unknown training data of closed-data models. We conclude that the strength of linear representations in LMs contains signal about the models' pretraining corpora that may provide new avenues for controlling and improving model behavior: particularly, manipulating the models' training data to meet specific frequency thresholds. We release our code to support future work.[1]

## 1    INTRODUCTION

Understanding how the content of pretraining data affects language model (LM) behaviors and performance is an active area of research (Ma et al., 2024; Xie et al., 2023; Aryabumi et al., 2025; Longpre et al., 2024; Wang et al., 2025; Seshadri et al., 2024; Razeghi et al., 2023; Wang et al., 2024). For instance, it has been shown that for specific tasks, models perform better on instances containing higher frequency terms than lower frequency ones (Razeghi et al., 2022; Mallen et al., 2023; McCoy et al., 2024). However, the ways in which frequency affects the internal representations of LMs to cause this difference in performance remain unclear. We connect dataset statistics to recent work in interpretability, which focuses on the emergence of simple linear representations of factual relations in LMs Hernandez et al. (2024); Chanin et al. (2024). Our findings demonstrate a strong correlation between these linear representations and the frequency of terms in the pretraining corpus.

---

[1]Code is available at https://github.com/allenai/freq, and for efficient batch search at https://github.com/allenai/batchsearch.

Linear representations in LMs have become central to interpretability research in recent years (Ravfogel et al., 2020; Elazar et al., 2021; Elhage et al., 2021; Slobodkin et al., 2023; Olah et al., 2020; Park et al., 2024; Jiang et al., 2024; Black et al., 2022; Chanin et al., 2024). Linear representations are essentially linear approximations (linear transforms, directions in space) that are simple to understand, and strongly approximate the complex non-linear transformations that networks are implementing. These representations are crucial because they allow us to localize much of the behavior and capabilities of LMs to specific directions in activation space. This allows for simple interventions to control model behaviors, i.e., steering (Todd et al., 2024; Subramani et al., 2022; Hendel et al., 2023; Rimsky et al., 2024).

Recent work by Hernandez et al. (2024) and Chanin et al. (2024) highlight how the linearity of different types of relations varies greatly depending on the specific relationships being depicted. For example, over 80% of entities in the "*country-largest-city*" relation, but less than 30% of entities in the "*star-in-constellation*" relation can be approximated this way (Hernandez et al., 2024). Such findings complicate the understanding of the Linear Representation Hypothesis, which proposes that LMs will represent features linearly (Park et al., 2024) without providing when/why these form. While Jiang et al. (2024) provide both theoretical and empirical evidence that the training objectives of LMs implicitly encourage linear representations, it remains unclear why some features are represented this way while others are not. This open question is a central focus of our investigation.

Whether linear representations for "common" concepts are more prevalent in models or simply easier to identify (using current methods) than those for less common concepts remains unclear. We hypothesize that factual relations exhibiting linear representations are correlated with higher mention frequencies in the pretraining data (as has been shown with static embeddings, see Ethayarajh et al., 2019), which we confirm in Section 4. Our results also indicate that this can occur at any point in pretraining, as long as a certain average frequency is reached across subject-object pairs in a relation. In order to count the appearance of terms in data corpora throughout training, we develop an efficient tool for counting tokens in tokenized batches of text, which we release to support future work in this area. We also explore whether the presence of linear representations can provide insights into relation term frequency. In Section 5, we fit a regression model to predict the frequency of individual terms (such as "The Beatles") in the pretraining data, based on metrics measuring the presence of a linear representation for some relation. For example, how well a linear transformation approximates the internal computation of the "*lead-singer-of*" relation mapping "John Lennon" to "The Beatles" can tell us about the frequency of those terms in the pretraining corpus.

Our findings indicate that the predictive signal, although approximate, is much stronger than that encoded in log probabilities and task accuracies alone, allowing us to estimate the frequencies of held-out relations and terms within approximate ranges. Importantly, this regression model generalizes beyond the specific LM it was trained on without additional supervision. This provides a valuable foundation for analyzing the pretraining corpora of closed-data models with open weights.

To summarize, in this paper we show that:

1. The development of linear representations for factual recall relations in LMs is related to frequency as well as model size.

2. Linear representations form at predictable frequency thresholds during training, regardless of when this frequency threshold is met for the nouns in the relation. The formation of these representations also correlates strongly with recall accuracy.

3. Measuring the extent to which a relation is represented linearly in a model allows us to predict the approximate frequencies of individual terms in the pretraining corpus of that model, even when we do not have access to the model's training data.

4. We release a tool for accurately and efficiently searching through tokenized text to support future research on training data.

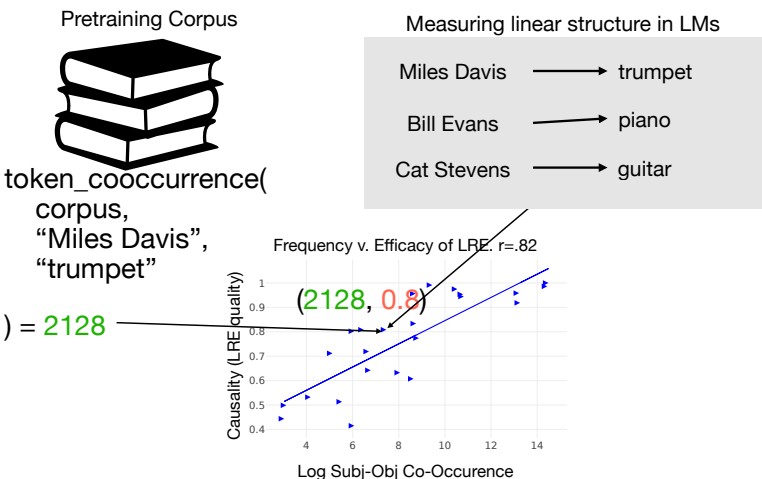

Figure 1: Overview of this work. Given a dataset of subject-relation-object factual relation triplets, we count subject-object co-occurrences throughout pretraining batches. We then measure how well the corresponding relations are represented within an LM across pretraining steps, using the Linear Relational Embeddings (LRE) method from Hernandez et al. (2024). We establish a strong relationship between average co-occurrence frequency and a model's tendency to form linear representations for relations. From this, we show that we can predict frequencies in the pretraining corpus

## 2 BACKGROUND

### 2.1 LINEAR REPRESENTATIONS

Vector space models have a long history in language processing, where geometric properties of these spaces were used to encode semantic information (Salton et al., 1975; Paccanaro & Hinton, 2001). When and why linear structure emerges without explicit bias has been of considerable interest since the era of static word embeddings. Work on skipgram models (Mikolov et al., 2013a) found that vector space models of language learn regularities which allow performing vector arithmetic between word embeddings to calculate semantic relationships (e.g., $\text{France} - \text{Paris} + \text{Spain} = \text{Madrid}$) (Mikolov et al., 2013b; Pennington et al., 2014). This property was subject to much debate, as it was not clear why word analogies would appear for some relations and not others (Köper et al., 2015; Karpinska et al., 2018; Gladkova et al., 2016). Followup work showed that linguistic regularities form in static embeddings for relations under specific dataset frequency constraints for relevant terms (Ethayarajh et al., 2019), but does not clearly relate to how modern LMs learn. More recently, there has been renewed interest in the presence of similar linear structure in models with contextual embeddings like transformer language models (Park et al., 2024; Jiang et al., 2024; Merullo et al., 2024). As a result, there are many ways to find and test for linear representations in modern LMs, though the relationship to pretraining data was not addressed (Huben et al., 2024; Gao et al., 2025; Templeton et al., 2024; Rimsky et al., 2024; Todd et al., 2024; Hendel et al., 2023; Hernandez et al., 2024; Chanin et al., 2024). Many of these share similarities in how they compute and test for linear representations. We focus on a particular class of linear representations called Linear Relational Embeddings (LREs) (Paccanaro & Hinton, 2001).

**Linear Relational Embeddings (LREs)** Hernandez et al. (2024) use a particular class of linear representation called a Linear Relational Embedding (Paccanaro & Hinton, 2001) to approximate the computation performed by a model to predict the objects that complete common `subject-relation-object` triplets as an affine transformation. This transform is calculated from a hidden state $\mathbf{s}$, the subject token representation at some middle layer of the model, to $\mathbf{o}$, the hidden state at the last token position and layer of the model (i.e., the final hidden state that decodes a token in an autoregressive transformer) within a natural language description of the relation. For example, given the input sequence "Miles Davis (`subject`) plays the (`relation`)", the goal is to approximate the computation of the object "trumpet", assuming the model predicts the object cor-

rectly. It was found that this transformation holds for nearly every subject and object in the relation set (such as "Cat Stevens plays the guitar") for some relations. This is surprising because, despite the nonlinearities within the many layers and token positions separating $\mathbf{s}$ and $\mathbf{o}$, a simple structure within the representation space well approximates the model's prediction process for a number of factual relations. In this work we study LREs under the same definition and experimental setup, because it allows us to predefine the concepts we want to search for (e.g., factual relations), as well as use a handful of representations to relate thousands of terms in the dataset by learning linear representations on a per-relation level.

Hernandez et al. calculate LREs to approximate an LM's computation as a first-order Taylor Series approximation. Let $F(\mathbf{s}, c) = \mathbf{o}$ be the forward pass through a model that produces object representation $\mathbf{o}$ given subject representation $\mathbf{s}$ and a few-shot context $c$, this computation is approximated as $F(\mathbf{s}, c) \approx W\mathbf{s} + b = F(\mathbf{s}_i, c) + W(\mathbf{s} - \mathbf{s}_i)$ where we approximate the relation about a specific subject $\mathbf{s}_i$. Hernandez et al. propose to compute $W$ and $b$ using the average of $n$ examples from the relation ($n = 8$ here) with $\frac{\partial F}{\partial \mathbf{s}}$ representing the Jacobian Matrix of $F$:

$$W = \mathbb{E}_{\mathbf{s}_i, c_i} \left[ \frac{\partial F}{\partial \mathbf{s}} \bigg|_{(\mathbf{s}_i, c_i)} \right] \quad \text{and} \quad b = \mathbb{E}_{\mathbf{s}_i, c_i} \left[ F(\mathbf{s}, c) - \frac{\partial F}{\partial \mathbf{s}} \mathbf{s} \bigg|_{(\mathbf{s}_i, c_i)} \right] \tag{1}$$

In practice, LREs are estimated using hidden states from LMs during the processing of the test example in a few-shot setup. For a relation like "*instrument-played-by–musician*", the model may see four examples (in the form "[X] plays the [Y]") and on the fifth example, when predicting e.g., "trumpet" from "Miles Davis plays the", the subject representation $\mathbf{s}$ and object representation $\mathbf{o}$ are extracted.

## 2.2 INFERRING TRAINING DATA FROM MODELS

There has been significant interest recently in understanding the extent to which it is possible to infer the training data of a fully trained neural network, including LMs, predominantly by performing membership inference attacks (Shokri et al., 2017; Carlini et al., 2022), judging memorization of text (Carlini et al., 2023; Oren et al., 2024; Shi et al., 2024), or inferring the distribution of data sources (Hayase et al., 2024; Ateniese et al., 2015; Suri & Evans, 2022). Our work is related in that we find hints of the pretraining data distribution in the model itself, but focus on how linear structure in the representations relates to training data statistics.

## 3 METHODS

Our analysis is twofold: counts of terms in the pretraining corpus of LMs, and measurements of how well factual relations are approximated by affine transformations. We use the OLMo model v1.7 (0424 7B and 0724 1B) (Groeneveld et al., 2024) and GPT-J (6B) (Wang & Komatsuzaki, 2021) and their corresponding datasets: Dolma (Soldaini et al., 2024) and the Pile (Gao et al., 2020), respectively. To understand how these features form over training time, we test eight model checkpoints throughout training in the OLMo family of models (Groeneveld et al., 2024).

### 3.1 LINEAR RELATIONAL EMBEDDINGS (LRES) IN LMS

We use a subset of the RELATIONS dataset Hernandez et al. (2024), focusing on the 25 factual relations of the dataset, such as *capital-city* and *person-mother* (complete list in Appendix B).[2] Across these relations, there are 10,488 unique subjects and objects. Following Hernandez et al. (2024), we fit an LRE for each relation on 8 examples from that relation, each with a 5-shot prompt. We use the approach from this work as described in Section 2.1.

---

[2]For the analysis, we drop "*landmark-on-continent*" because 74% of the answers are Antarctica, making it potentially confounding for extracting a representation for the underlying relation. Factual relations are much easier to get accurate counts for, so we leave non-factual relations for future work (e.g., although LMs associate the "pilot" occupation with men, this relation does not map to the word "man" the way "France" maps to "Paris"; see §3.2).

**Fitting LREs**  Hernandez et al. (2024) find that Equation 1 underestimates the optimal slope of the linear transformation, so they scale each relation's $W$ by a scalar hyperparameter $\beta$. Unlike the original work, which finds one $\beta$ per model, we use one $\beta$ per relation, as this avoids disadvantaging specific relations. Another difference in our calculation of LREs is that we do not impose the constraint that the model has to predict the answer correctly to be used as one of the 8 examples used to approximate the Jacobian Matrix. Interestingly, using examples that models predict incorrectly to fit Equation 1 works as well as using only correct examples. We opt to use this variant as it allows us to compare different checkpoints and models (§4) with linear transformations trained on the same 8 examples, despite the fact that the models make different predictions on these instances. We explore the effect of example choice in Appendix B and find that it does not make a significant difference. We also explore the choice of layer in Appendix C.

**Metrics**  To evaluate the quality of LREs, Hernandez et al. (2024) introduce two metrics that measure the quality of the learned transformations. **Faithfulness** measures whether the transformation learned by the LRE produces the same object token prediction as the original LM. **Causality** measures the proportion of the time a prediction of an object can be changed to the output of a different example from the relation (e.g., editing the *Miles Davis* subject representation so that the LM predicts he plays the *guitar*, instead of the *trumpet*). For specifics on implementation, we refer the reader to Hernandez et al. (2024). We consider an LRE to be high 'quality' when it scores highly on these metrics, as this measures when an LRE works across subject-object pairs within the relation. In general, we prefer to use causality in our analysis, as faithfulness can be high when LMs predict the same token very often (like in early checkpoints).

## 3.2  Counting Frequencies Throughout Training

A key question we explore is how term frequencies affect the formation of linear representations. We hypothesize that more commonly occurring relations will lead to higher quality LREs for those relations. Following Elsahar et al. (2018); Elazar et al. (2022), we count an occurrence of a relation when a subject and object co-occur together. While term co-occurrence is used as a proxy for the frequency of the entire triplet mentioned in text, Elsahar et al. (2018) show that this approximation is quite accurate. We now discuss how to compute these co-occurrence counts.

**What's in My Big Data? (WIMBD)**  Elazar et al. (2024) index many popular pretraining datasets, including Dolma (Soldaini et al., 2024) and the Pile (Gao et al., 2020), and provide search tools that allow for counting individual terms and co-occurrences within documents. However, this only gives us counts for the full dataset. Since we are interested in counting term frequencies throughout pretraining, we count these within training batches of OLMo instead. When per-batch counts are not available, WIMBD offers a good approximation for final checkpoints, which is what we do in the case of GPT-J. We compare WIMBD co-occurrence counts to the Batch Search method (described below) for the final checkpoint of OLMo in Appendix D, and find that the counts are extremely close: The slope of the best fit line for BatchCount against WIMBDCount is .94, because co-occurrence counts are overestimated when considering the whole document.

**Batch Search**  Data counting tools cannot typically provide accurate counts for model checkpoints at arbitrary training steps. Thus, we design a tool to efficiently count exact co-occurrences within sequences of tokenized batches. This also gives us the advantage of counting in a way that is highly accurate to how LMs are trained; since LMs are trained on batches of fixed lengths which often split documents into multiple sequences, miscounts may occur unless using tokenized sequences. Using this method, we note every time one of our 10k terms appears throughout a dataset used to pretrain an LM. We count a co-occurrence as any time two terms appear in the same sequence within a batch (a $(\text{batch-size}, \text{sequence-length})$ array). We search 10k terms in the approximately 2T tokens of Dolma (Soldaini et al., 2024) this way. Using our implementation, we are able to complete this on 900 CPUs in about a day. To support future work, we release our code as Cython bindings that integrate out of the box with existing libraries.

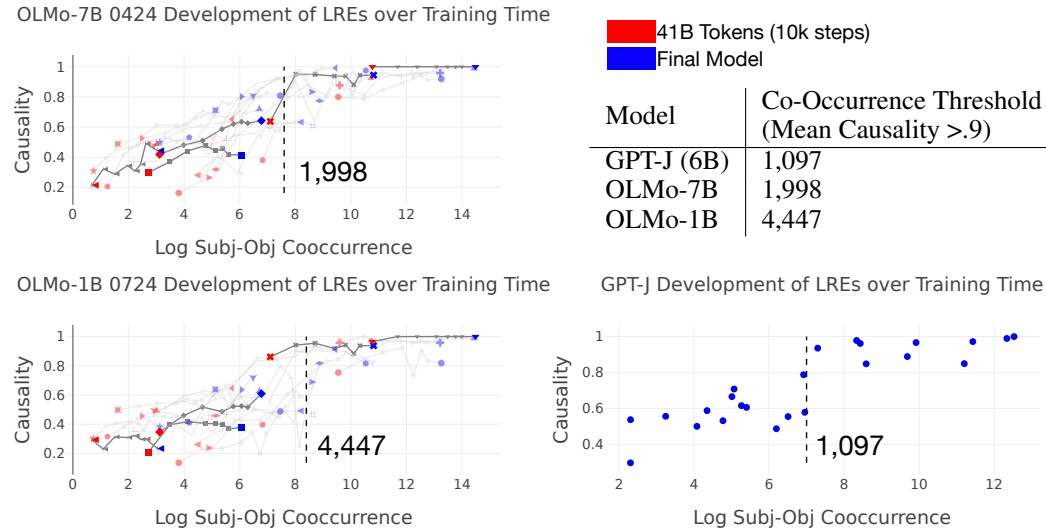

Figure 2: We find that LREs have consistently high causality scores across relations after some average frequency threshold is reached (table, top right). In OLMo models, red dots show the model's LRE performance at 41B tokens, and blue dots show the final checkpoint performance ( 550k steps in 7B). Gray dots show intermediate checkpoints. We highlight Even at very early training steps, if the average subject-object cooc. count is high enough, the models are very likely to already have robust LREs formed in the representation space. Symbols represent different relations. Highlighted relations are shown in darker lines.[5]

## 4 FREQUENCY OF SUBJECT-OBJECT CO-OCCURRENCES ALIGNS WITH EMERGENCE OF LINEAR REPRESENTATIONS

In this section, we explore when LREs begin to appear at training time and how these are related to pretraining term frequencies. Our main findings are that (1) average co-occurrence frequency within a relation strongly correlates with whether an LRE will form; (2) the frequency effect is independent of the pretraining stage; if the average subject-object co-occurrence for a relation surpasses some threshold, it is very likely to have a high-quality LRE, even for early pretraining steps.

### 4.1 SETUP

Using the factual recall relations from the Hernandez et al. (2024) dataset, we use the Batch Search method (§3.2) to count subject and object co-occurrences within sequences in Dolma (Soldaini et al., 2024) used to train the OLMo-1B (v. 0724) and 7B (v. 0424) models (Groeneveld et al., 2024). The OLMo family of models provides tools for accurately recreating the batches from Dolma, which allow us to reconstruct the data the way the model was trained. We also use GPT-J (Wang & Komatsuzaki, 2021) and the Pile (Gao et al., 2020) as its training data, but since we do not have access to accurate batches used to train it, we use WIMBD (Elazar et al., 2024) to count subject-object counts in the entire data. We fit LREs on each relation and model separately. Hyperparameter sweeps are in Appendix C. OLMo also releases intermediate checkpoints, which we use to track development over pretraining time. We use checkpoints that have seen {41B, 104B, 209B, 419B, 628B, 838B, 1T, and 2T} tokens.[3] We use the Pearson coefficient for measuring correlation.

### 4.2 RESULTS

Our results are summarized in Figure 2. We report training tokens because the step count differs between 7B and 1B. Co-occurrence frequencies highly correlate with causality ($r = 0.82$). This

---

[3]In OLMo-7B 0424, this corresponds to 10k, 25k, 50k, 100k, 150k, 200k, 250k, 409k pretraining steps

[5]These are: 'country largest city', 'country currency', 'company hq', 'company CEO', and 'star constellation name' in order from best to worst performing final checkpoints.

is notably higher than the correlations with subject frequencies: $r = 0.66$, and object frequencies: $r = 0.59$ for both OLMo-7B and OLMo-1B, respectively.

We consider a causality score above $0.9$ to be nearly perfectly linear. The table in Figure 2 shows the co-occurrence counts above which the average causality is above $0.9$ and is shown by dashed black lines on the scatterplots. Regardless of pretraining step, models that surpass this threshold have very high causality scores. Although we cannot draw conclusions from only three models, it is possible that scale also affects this threshold: OLMo-7B and GPT-J (6B params) require far less exposure than OLMo-1B.

### 4.3 RELATIONSHIP TO ACCURACY

Increased frequency (or a proxy for it) was shown to lead to better factual recall in LMs (Chang et al., 2024; Mallen et al., 2023). However, it remains unknown whether high accuracy entails the existence of a linear relationship. Such a finding would inform when we expect an LM to achieve high accuracy on a task. We find that the correlation between causality and subject-object frequency is higher than with 5-shot accuracy ($0.82$ v.s. $0.74$ in OLMo-7B), though both are clearly high. In addition, there are a few examples of high accuracy relations that do not form single consistent LREs. These relations are typically low frequency, such as star constellation name, which has 84% 5-shot accuracy but only 44% causality (OLMo-7B), with subjects and objects only co-occurring about 21 times on average across the full dataset. In general, few-shot accuracy closely tracks causality, consistent with arguments that in-context learning allows models to identify linear mappings between input-output pairs (Hendel et al., 2023; Garg et al., 2022). We find that causality increases first in some cases, like "*food-from-country*" having a causality of 65% but a 5-shot accuracy of only 42%. This gap is consistently closed through training. In the final model, causality and 5-shot accuracy are within 11% on average. We report the relationship between every relation, zero-shot, and few-shot accuracy for OLMo models across training in Appendix F.

A fundamental question in the interpretability community is under what circumstances linear structures form. While previous work has shown that the training objective encourages this type of representation (Jiang et al., 2024), our results suggest that the reason why some concepts form a linear representation while others do not is strongly related to the pretraining frequency.

## 5 LINEAR REPRESENTATIONS HELP PREDICT PRETRAINING CORPUS FREQUENCIES

In this section, we aim to understand this relationship further by exploring what we can understand about pretraining term frequency from linearity of LM representations. We target the challenging problem of predicting how often a term, or co-occurrence of terms, appears in an LM's training data from the representations alone. Such prediction model can be useful, if it generalizes, when applied to other models whose weights are open, but the data is closed. For instance, such predictive model could tell us whether a model was trained on specific domains (e.g., Java code) by measuring the presence of relevant LREs. First, we show that LRE features encode information about frequency that is not present using probabilities alone. Then, we show how a regression fit on one model generalizes to the features extracted from another without any information about the new model's counts.

### 5.1 EXPERIMENTAL SETUP

We fit a regression to the Relations dataset (Hernandez et al., 2024) using OLMo-7B LRE features and log probabilities. We fit 24 models such that each relation is held out once per random seed across 4 seeds. We train a random forest regression model with 100 decision tree estimators to predict the frequency of terms (either the subject-object frequency, or the object frequency alone; e.g., predicting "John Lennon" and "The Beatles" or just "The Beatles") from one of two sets of features. Our baseline set of features is based on likelihood of recalling a fact. Given some few-shot context from the relations dataset ("John Lennon is a lead singer of") we extract the log probability of the correct answer, as well as the average accuracy on this prompt across 5 trials. The intuition is that models will be more confident about highly frequent terms. The other set of features include the first, as well as faithfulness and causality measurement.

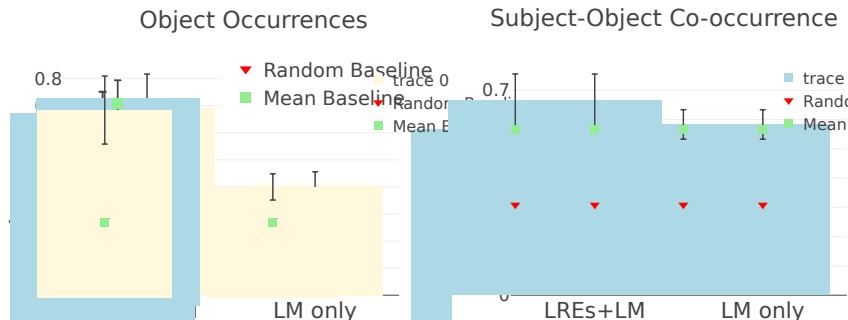

Figure 3: Within-Magnitude accuracy (aka the proportion of predictions within one order of magnitude of ground truth) for models predicting object and subject-object co-occurrences in heldout relations. Using LRE features outperforms LM only features by about 30%. We find that it is much easier to predict object frequencies; the subj-obj. prediction models with LRE features only marginally outperform baseline performance.

We use Faithfulness and Causality as defined in Hernandez et al. (2024) as well as two other metrics: **Faith Prob.**, which is the log probability of the correct answer as produced by an LRE, and **Hard Causality**, which is the same as the "soft" variant, but only counts the proportion of times the causality edit produces the target answer as the number one prediction. We use every example from the relations for which there are more than one object occurrence or subject-object co-occurrence. We do not provide an explicit signal for which relation an example comes from, but due to the bias of subjects/objects having similar frequencies within a relation, we train multiple models and evaluate on held out relations and average performance. In all settings, the held out set objects and relations are guaranteed to not have been in the training set.

## 5.2 LRE METRICS ENCODE FINE-GRAINED FREQUENCY INFORMATION

Because of the difficulty of predicting the exact number of occurrences, we report accuracy within one order of magnitude of the ground truth. This measures whether the predicted value is within a reasonable range of the actual value. Results are shown in Figure 3. We find that language modeling features do not provide any meaningful signal towards predicting object or subject-object frequencies, and are only marginally above the baseline of predicting the average or random frequencies from the training data. On object frequency predictions, we find that LRE features encode a strong signal allowing for accurate predictions about 70% of the time. Mean absolute error of the predictions (in natural log space) for LRE features (LM-only features) are 2.1, (4.2) and 1.9, (2.3) on object prediction and subject-object prediction tasks, respectively. We find that subject-object co-occurrence frequency is likely too difficult to predict given the signals that we have here, as our predictions are higher than, but within one standard deviation of the mean baseline.

**Feature Importance:** How important are LRE features for predicting the frequency of an item? We perform feature permutation tests to see how much each feature (LRE features and log probs) contributes to the final answer. First, we check to see which features used to fit the regression are correlated, as if they are, then perturbing one will leave the signal present in another. In Appendix E, we show that only faithfulness and faith probability are strongly correlated, so for this test only, we train models with a single PCA component representing 89% of the variance of those two features. We find that hard causality is by far the most important feature for generalization performance, causing a difference of about 15% accuracy, followed by faithfulness measures with 5% accuracy, providing evidence that the LRE features are encoding an important signal.

## 5.3 GENERALIZATION TO A NEW LM

Next, we test the ability to generalize the regression fit of one LM to another, without requiring further supervision. If such a model could generalize, we can predict term counts to models for which we do not have access to their pretraining data. We keep the objective the same and apply

Table 1: Within-Magnitude accuracy for different settings of train and test models. Overall, we find that fitting a regression on one model's LREs and evaluating on the other provides a meaningful signal compared to fitting using only log probability and task performance, or predicting the average training data frequency. The metric here is proportion of predictions within one order of 10x the ground truth. Here, Eval. on GPT-J means the regression is fit on OLMo and evaluated on GPT-J.

| Model | Predicting Object Occs. | | Predicting Subject-Object Co-Occs. | |
|---|---|---|---|---|
| | Eval. on GPT-J | Eval. on OLMo | Eval. on GPT-J | Eval. on OLMo |
| LRE Features | 0.65±0.12 | 0.49±0.12 | 0.76±0.12 | 0.68±0.08 |
| LogProb Features | 0.42±0.10 | 0.41±0.09 | 0.66±0.09 | 0.60±0.07 |
| Mean Freq. Baseline | 0.31±0.15 | 0.41±0.17 | 0.57±0.15 | 0.67±0.16 |

the regression model, fit for example on OLMo ("Train OLMo" setting), to features extracted from GPT-J, using ground truth counts from The Pile (and vice versa, i.e., the "Train GPT-J" setting).

We again train a random forest regression model to predict the frequency of terms (either the subject-object frequency, or the object frequency alone; e.g., predicting "John Lennon" and "The Beatles" or just "The Beatles") on features from one of two models: either OLMo-7B (final checkpoint) or GPT-J, treating the other as the 'closed' model. We test the hypothesis that LRE features (**faithfulness, causality**) are useful in predicting term frequencies across different models, with the hope that this could be applied to dataset inference methods in the future, where access to the ground truth pretraining data counts is limited or unavailable.

**Results** Our results are presented in Table 1. First, we find that there is a signal in the LRE features that does not exist in the log probability features: We are able to fit a much better generalizable model when using LRE features as opposed to the LM probabilities alone. Second, evaluating on the LRE features of a heldout model (scaled by the ratio of total tokens trained between the two models) maintains around the same accuracy when fit on exact counts from OLMo, allowing us to predict occurrences without access to the GPT-J pretraining data. We find that predicting either the subject-object co-occurrences or object frequencies using LREs alone is barely better than the baseline. This task is much more difficult than predicting the frequency of the object alone, but our model may just also be unable to account for outliers in the data, which is tightly clustered around the mean (thus giving the high mean baseline performance of between approx. 60-70%). Nevertheless, we show that linear structure for relations within LM representations encode a rich signal representing dataset frequency.

## 5.4 Error Analysis

In Table 2 we show example predictions from our regression model that we fit on OLMo and evaluate on heldout relations with LREs measured on GPT-J. We find that some relations transfer more easily than others, with the star constellation name transferring especially poorly. In general, the regression transfers well, without performance deteriorating much (about 5% accuracy: see Figure 3 compared to the evaluation of GPT-J in Table 1), suggesting LREs encode information in a consistent way across models. We also find that the regression makes use of the full prediction range, producing values in the millions (see Table 2) as well as in the tens; The same regression shown in the table also predicts 59 occurrences for "Caroline Bright" (Will Smith's mother) where the ground truth is 48.

## 6 Discussion

**Connection to Factual Recall** Work in interpretability has focused largely around linear representations in recent years, and our work aims to address the open question of the conditions in which they form. We find that coherent linear representations form when the relevant terms (in this case subject-object co-occurrences) appear in pretraining at a consistent enough rate. Analogously, Chang et al. (2024) show that repeated exposure encourages higher retention of facts. Future work could investigate the connection between factual recall accuracy and linear representations.

Table 2: Examples of a regression fit on OLMo LRE metrics and evaluated on GPT-J on heldout relations, demonstrating common error patterns: 1. Predictions are better for relations that are closer to those found in fitting the relation (country related relations), 2. Some relations, like star-constellation perform very poorly, possibly due to low frequency, 3. the regression model can be sensitive to the choice of subject (e.g., William vs. Harry), telling us the choice of data to measure LREs for is important for predictions.

| Predicting Object Frequency in GPT-J, Regression fit on OLMo | | | | | |
|---|---|---|---|---|---|
| **Relation** | **Subject** | **Object** | **Prediction** | **Ground Truth** | **Error** |
| *landmark-in-country* | Menangle Park | Australia | 2,986,989 | 3,582,602 | 1.2x |
| *country-language* | Brazil | Portuguese | 845,406 | 561,005 | 1x |
| *star-constellation name* | Arcturus | Boötes | 974,550 | 2,817 | 346x |
| *person-mother* | Prince William | Princess Diana | 5,826 | 27,094 | 4.6x |
| *person-mother* | Prince Harry | Princess Diana | 131 | 27,094 | 207x |

**Linear Representations in LMs** The difficulty of disentangling the formation of linear representations from increases in relation accuracy, especially in the few-shot case, is interesting. Across 24 relations, only the "*star-constellation-name*" and "*product-by-company*" relations have few-shot accuracies that far exceed their causality scores (and both are low frequency). Thus, it is still an open question how LMs are able to recall these tasks. The fact that few-shot accuracy and causality seem so closely linked is consistent with findings that ICL involves locating the right task (Min et al., 2022) and applying a 'function' to map input examples to outputs (Hendel et al., 2023; Todd et al., 2024). The finding that frequency controls this ability is perhaps unsurprising, as frequency also controls this linear structure emerging in static embeddings (Ethayarajh et al., 2019). Jiang et al. (2024) prove a strong frequency-based condition (based on matched log-odds between subjects and objects) and an implicit bias of gradient descent (when the frequency condition is not met) encourage linearity in LLMs; our work empirically shows conditions where linear representations tend to form in more realistic settings. If LMs are 'only' solving factual recall or performing ICL through linear structures, it is surprising how well this works at scale, but the simplicity also provides a promising way to understand LMs and ICL in general. An interesting avenue for future work would be to understand if and when LMs use a method that is not well approximated linearly to solve these types of tasks, as recent work has shown non-linearity can be preferred for some tasks in recurrent networks (Csordás et al., 2024).

**Future Work in Predicting Dataset Frequency** The ability to predict the contents of pretraining data is an important area for investigating memorization, contamination, and privacy of information used to train models. In our approach, we show it is possible to extract pretraining data signal without direct supervision. Without interpretability work on the nature of representations in LMs, we would not know of this implicit dataset signal, and we argue that interpretability can generate useful insights more broadly as well. Extensions on this work could include more information to tighten the prediction bounds on frequency, such as extracting additional features from the tokenizer (Hayase et al., 2024). We hope this work encourages future research in other ways properties of pretraining data affect LM representations for both improving and better understanding these models.

## 7 CONCLUSION

We find a connection between linear representations of subject-relation-object factual triplets in LMs and the pretraining frequencies of the subjects and objects in those relations. This finding can guide future interpretability work in deciphering whether a linear representation for a given concept will exist in a model, since we observe that frequencies below a certain threshold for a given model will not yield LREs (a particular class of linear representation). From there we show that we can use the presence of linear representations to predict with some accuracy the frequency of terms in the pretraining corpus of an open-weights, closed-data model without supervision. Future work could aim to improve on our bounds of predicted frequencies. Overall, our work presents a meaningful step towards understanding the interactions between pretraining data and internal LM representations.

## ACKNOWLEDGMENTS

This work was performed while JM was an intern at Ai2. We thank the anonymous reviewers and members of the Aristo and AllenNLP teams at Ai2 for valuable feedback.

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

## A    LIMITATIONS

While our approach thoroughly tracks exposure to individual terms and formation of LRE features across pretraining, we can not draw causal[6] claims about how exposure affects individual representations, due to the cost of counterfactual pretraining. We try to address this by showing the frequency of individual terms can be predicted with some accuracy from measurements of LRE presence. We motivate this approach as a possible way to detect the training data of closed-data LMs; however, we are not able to make any guarantees on its efficacy in settings not shown here, and would caution drawing strong conclusions without additional information. Furthermore, we find that our method is relatively worse at predicting subject-object co-occurrences than object occurrences, and our method fails to account for the harder task. Future work could expand on this tool by incorporating it with other data inference methods for greater confidence. We also do not discuss the role of the presentation of facts on the formation of LRE features, but following Elsahar et al. (2018) and the strength of the relationship we find, we speculate this has minimal impact. Note that the BatchSearch tool we release tracks the exact position index of the searched terms, thus facilitating future work on questions about templates and presentation of information.

## B    EFFECT OF TRAINING ON INCORRECT EXAMPLES

In Hernandez et al. (2024), examples are filtered to ones in which the LM gets correct, assuming that an LRE will only exist once a model has attained the knowledge to answer the relation accuracy (e.g., knowing many country capitals). We find that the choice of examples for fitting LREs is not entirely dependent on the model 'knowing' that relation perfectly (i.e., attains high accuracy). This is convenient for our study, where we test early checkpoint models, that do not necessarily have all of the information that they will have seen later in training. In Figure 5, we show faithfulness on relations where the LRE was fit with all, half, or zero correct examples. We omit data for which the model did not get enough incorrect examples. Averages across relations for which we have enough data are shown in Figure 4, which shows that there is not a considerable difference in the choice of LRE samples to train with.

## C    LRE HYPERPARAMETER TUNING

There are three hyperparameters for fitting LREs: **layer** at which to edit the subject, the **beta** term used to scale the LRE weight matrix, and the **rank** of the pseudoinverse matrix used to make edits for

---

[6]And thus mechanistic, in the narrow technical sense of the term (Saphra & Wiegreffe, 2024).

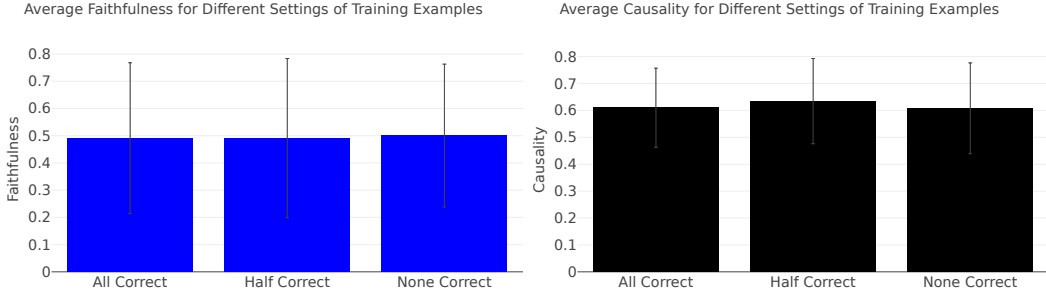

Figure 4: Average Causality and Faithfulness results across relations depending on if the LRE was fit with correct or incorrect samples. We find no notable difference in the choice of examples.

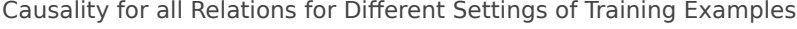

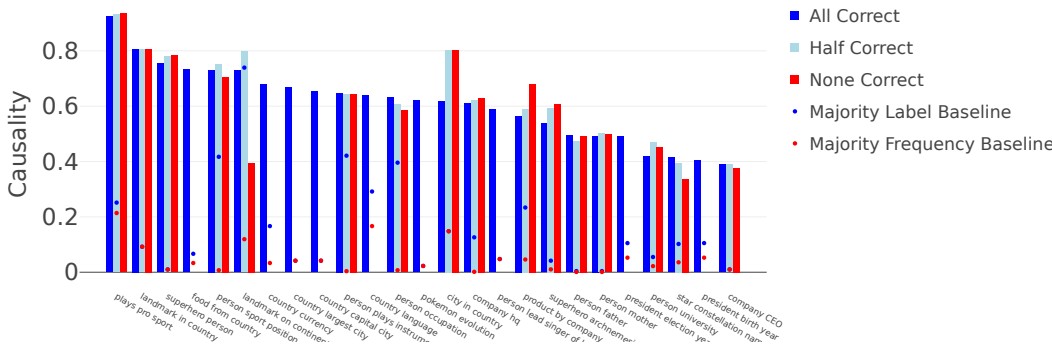

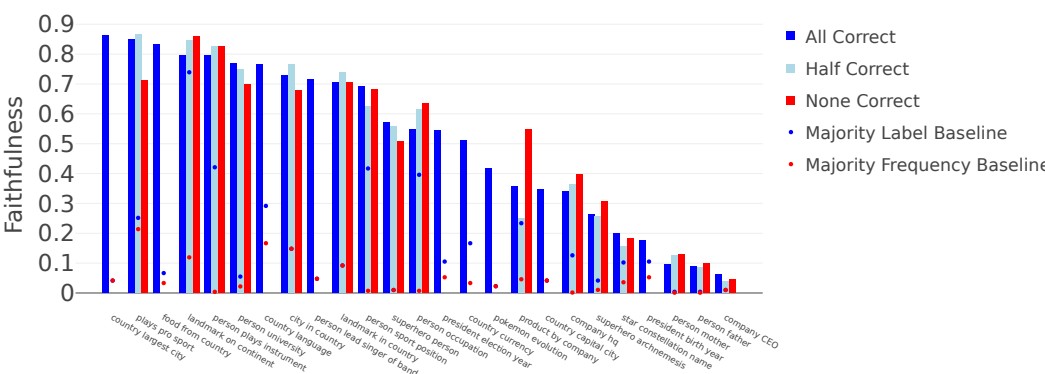

Figure 5: Causality and Faithfulness results for each relation depending on if the LRE was fit with correct or incorrect samples. Note that relations with only one bar do not have zeros in the other categories. It means that there was not enough data that the model (OLMo-7B) got wrong to have enough examples to fit.

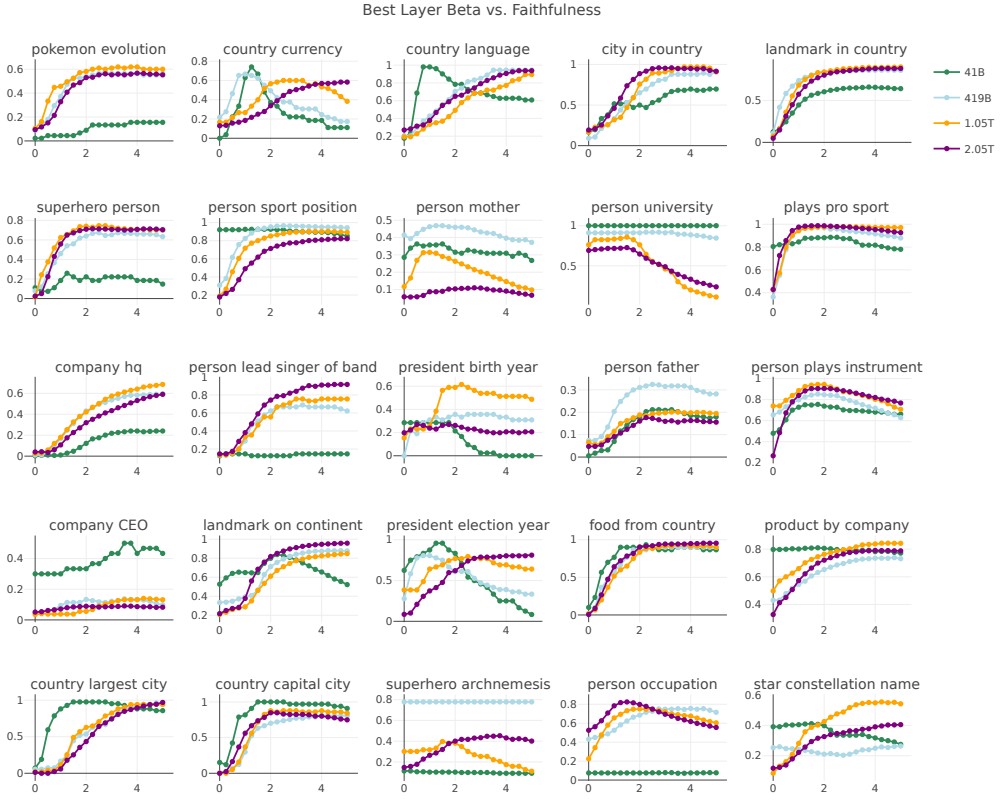

Figure 6: OLMo 0424 7B per layer faithfulness scores as a function of the choice of layer at which to fit the LRE. Note we do not use these results to choose the layer for the LRE, instead preferring the results from the causality sweep.

measuring causality. Beta is exclusive to measuring faithfulness and rank is exclusive to causality. We test the same ranges for each as in Hernandez et al. (2024): [0, 5] beta and [0, full_rank] for causality at varying intervals. Those intervals are every 2 from [0,100], every 5 from [100,200], every 25 from [200, 500], every 50 from [500, 1000], every 250 from [1000, hidden_size]. We perform the hyperparameter sweeps across faithfulness and causality, but we choose the layer to edit based on the causality score. In cases where this is not the same layer as what faithfulness would decide, we use the layer causality chooses, as it would not make sense to train one LRE for each metric. We refer the reader to Hernandez et al. (2024) for more details on the interactions between hyperparameters and the choice of layer. The results of our sweeps on OLMo-7B across layers in Figures 6 and 7 and across beta and rank choices in Figures 8 and 9.

## D    BATCH SEARCH COUNTS COMPARED TO WIMBD

In Figure 10, we find that What's in My Big Data (Elazar et al., 2024) matches very well to batch search co-occurrences; however, WIMBD tends to over-predict co-occurrences (slope less than 1), due to the sequence length being shorter than many documents, as discussed in the main paper.

## E    FEATURE CORRELATIONS AND IMPORTANCES

Our feature importance test is shown in Figure 12. This permutation test was done on the heldout data to show which features contribute the most to generalization performance. We use PCA to

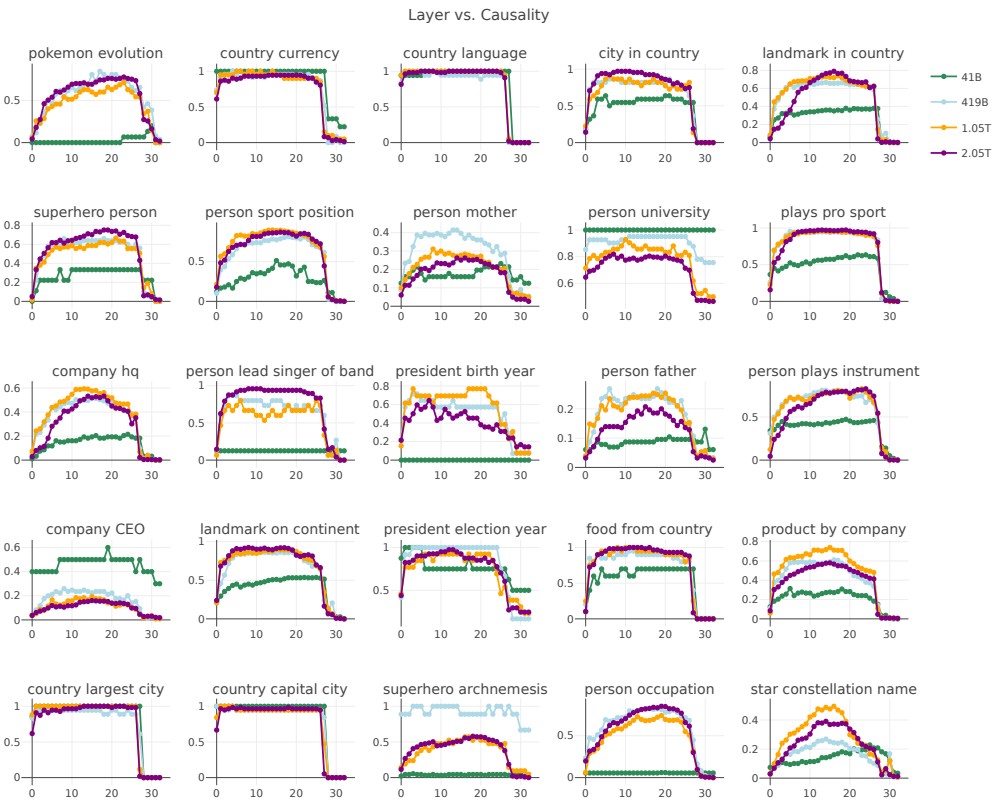

Figure 7: OLMo 0424 7B per layer causality scores as a function of the choice of layer at which to fit the LRE.

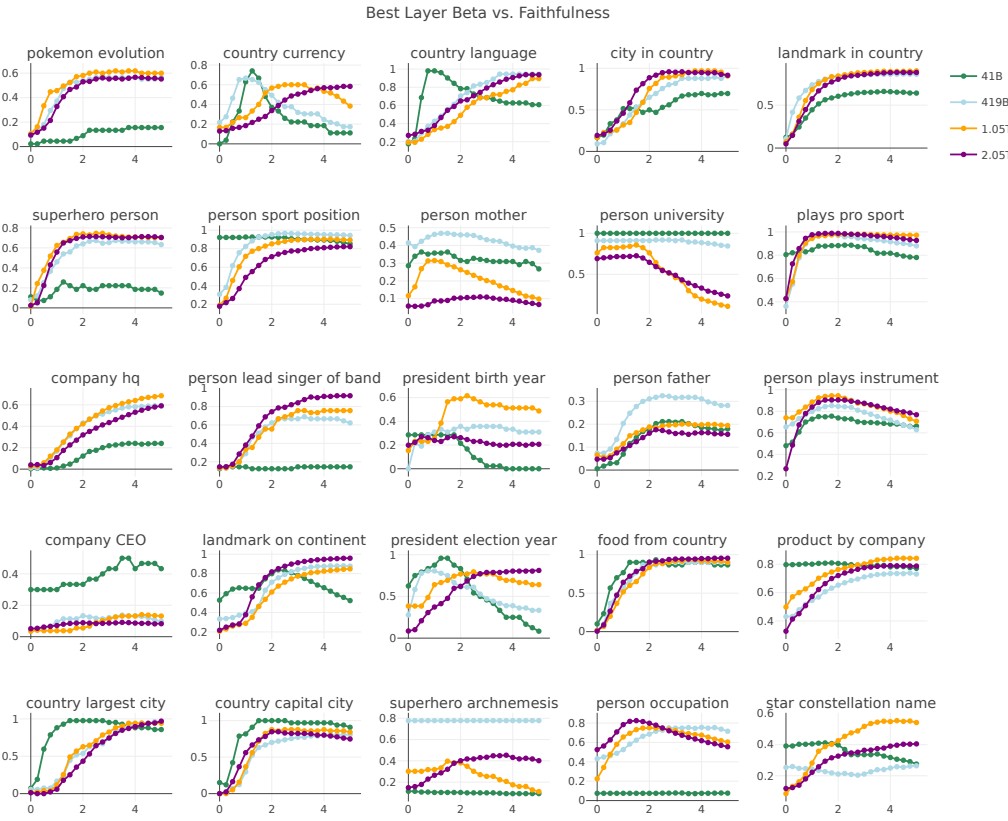

Figure 8: OLMo 0424 7B LRE Beta hyperparameter sweep at highest performing layer.

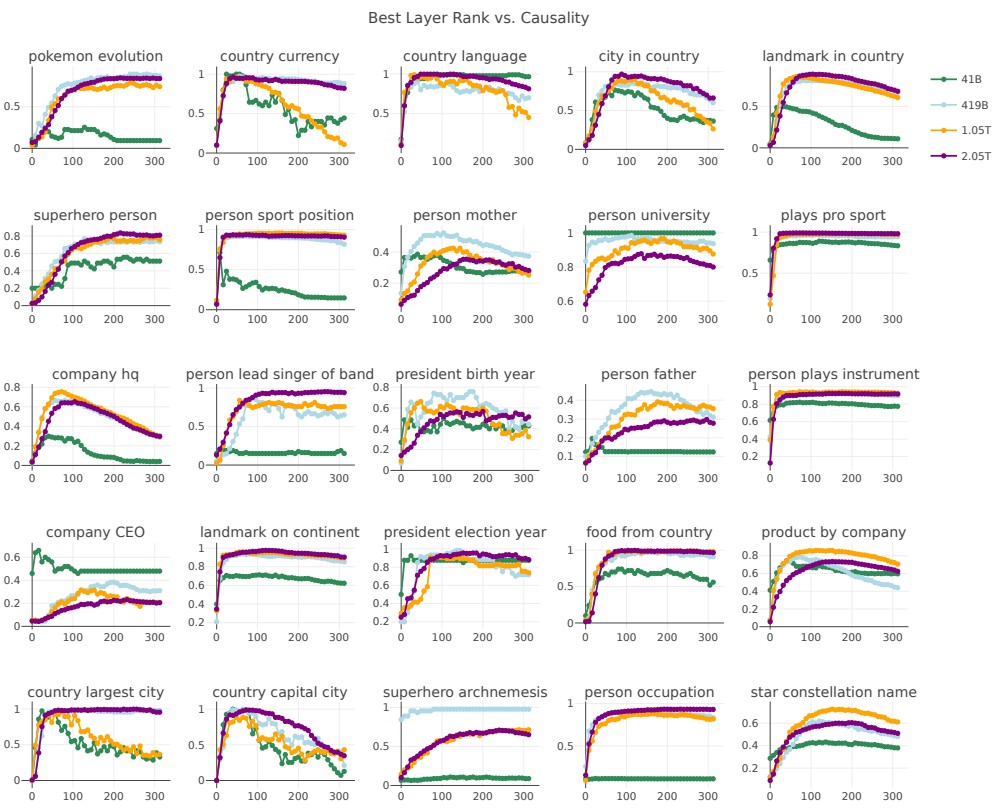

Figure 9: OLMo 0424 7B LRE Rank hyperparameter sweep at highest performing layer.

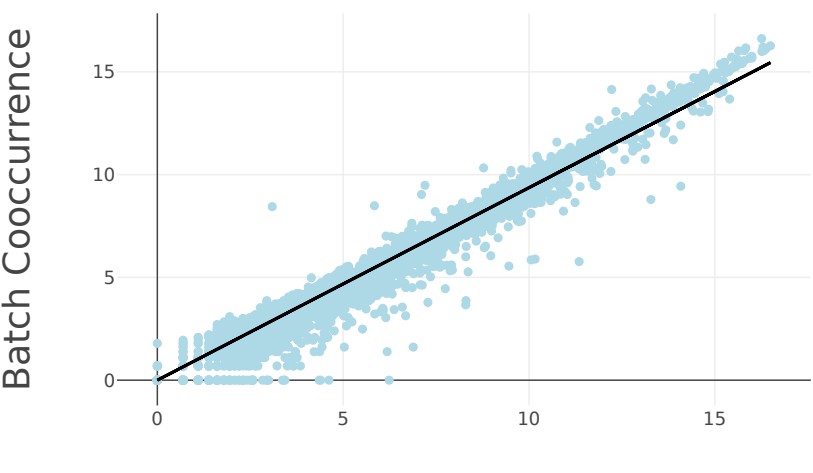

Figure 10: Comparison between WIMBD and Batch Search subject-object co-occurrences

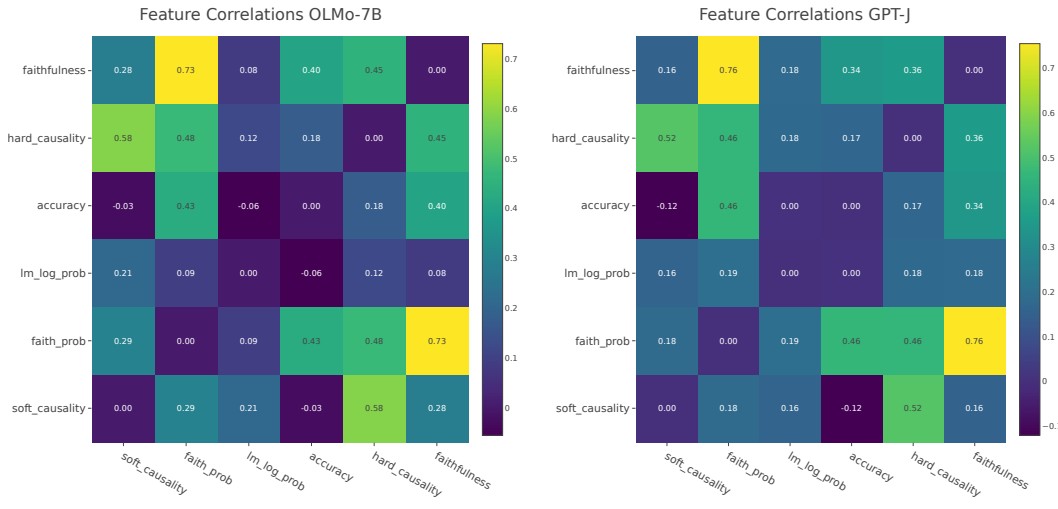

Figure 11: Correlations between each feature in our regression analysis. Because of the high correlation between faithfulness metrics, we use a single dimensional PCA to attain one feature that captures 89% of the variance of both for the purposes of doing feature importance tests. Note that we zero out the diagonal (which has values of 1) for readability.

reduce the faithfulness features to one feature for the purposes of this test. Correlations are shown in Figure 11

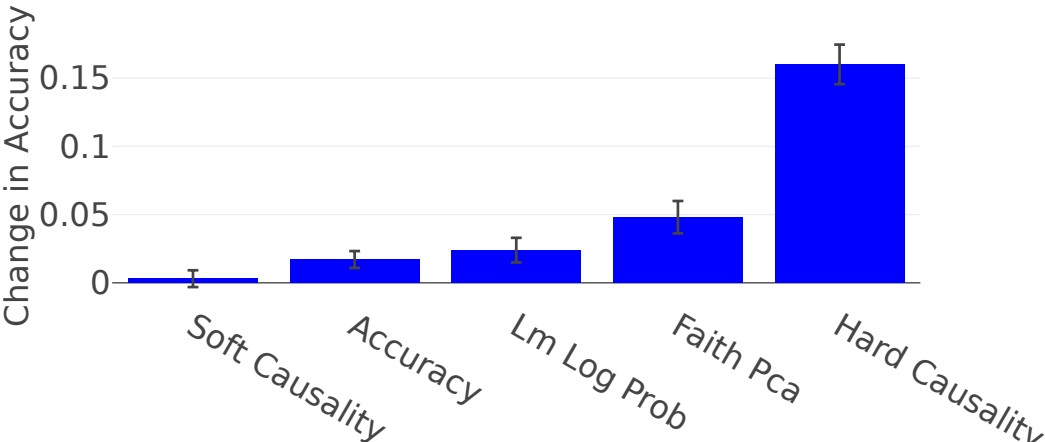

Figure 12: Hard causality is by far the most important feature for generalizing to new relations when predicting Object frequencies, causing a change in about 15% accuracy.

## F    RELATIONSHIP BETWEEN CAUSALITY AND ACCURACY

In this section, we provide more detail on the relationship between the formation of linear representations and accuracy on in-context learning tasks. Although the two are very highly correlated, we argue that accuracy and LRE formation are somewhat independent.

We show this relationship across training for OLMo-1B in Figure 13 and 7B in Figure 14.

## G    EXTENDING TO COMMONSENSE RELATIONS

Following Elsahar et al. (2018), we focus on factual relations because subject-object co-occurrences are shown to be a good proxy for mentions of the fact. For completeness, we consider 8 additional commonsense relations here. Results for OLMo-7B are shown in Figure 15. We show that frequency is correlated with causality score (.42) in these cases as well, but it is possible subject-object frequencies do not accurately track occurrences of the relation being mentioned. For example, in the "task person type" relation, the co-occurrence count of the subject "researching history" and the object "historian" does not convincingly describe all instances where the historian concept is defined during pretraining. Co-occurrences are perhaps more convincingly related to how a model learns that the outside of a coconut is brown, however (the fruit outside color relation). Therefore, we caution treating these under the same lens as the factual relations. Nevertheless, we believe these results are an interesting perspective on how a different relation family compares to factual relations.

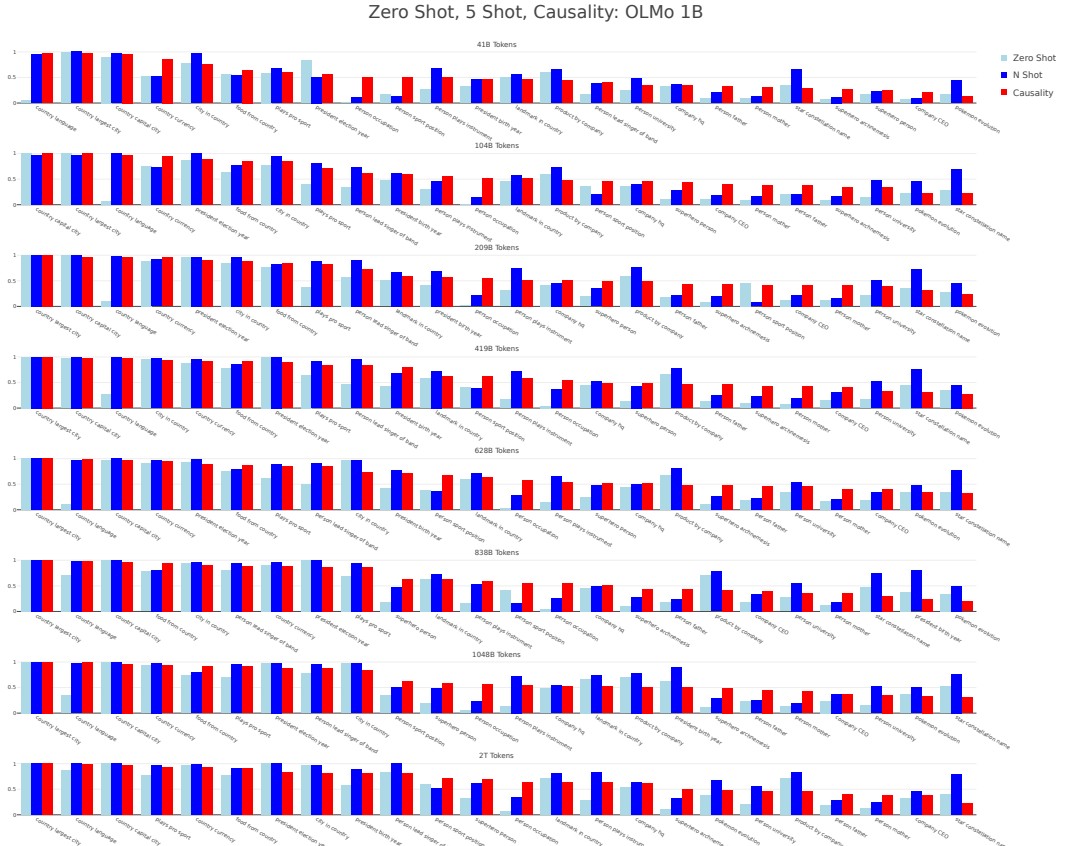

Figure 13: Zero shot, 5-shot accuracies against causality for each relation across training time in OLMo-1B

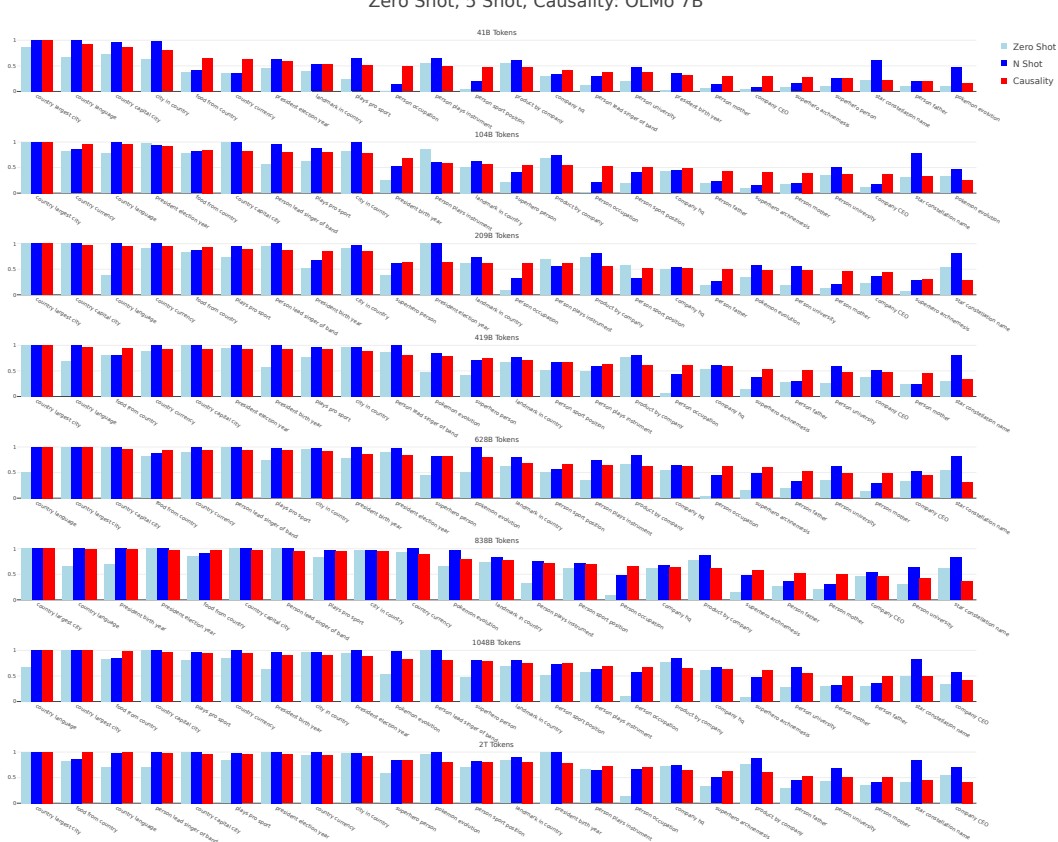

Figure 14: Zero shot, 5-shot accuracies against causality for each relation across training time in OLMo-7B

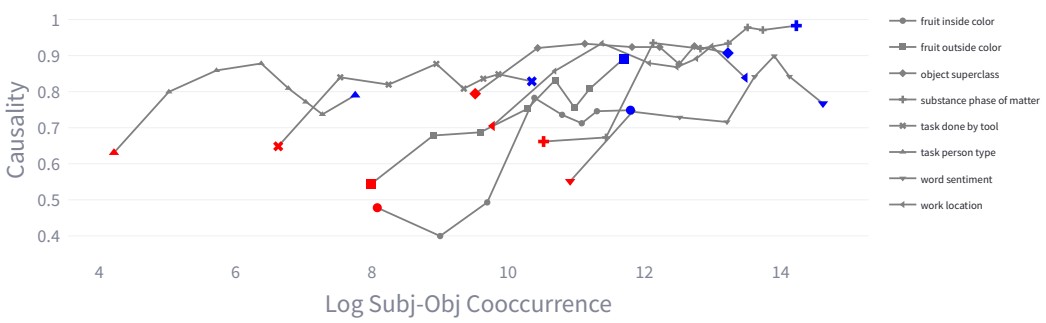

Figure 15: Commonsense relations compared to pretraining time in OLMo-7B.

