# OpenReview forum: "On Linear Representations and Pretraining Data Frequency in Language Models"
_ICLR.cc/2025/Conference — ICLR 2025 Poster_

### Official Review · Reviewer_juu2 · 2024-10-29

**Soundness:** 3
**Presentation:** 3
**Contribution:** 2
**Rating:** 6
**Confidence:** 4

**Summary:**

This paper finds that the formation of linear representations for factual recall relations in LMs is highly correlated with the frequency of subject-object cooccurrence in the pretraining data. The formation of linear representation can happen at any stage of pretraining as long as the subj-obj cooccurrence exceeds some threshold, i.e., a linear representation can form consistently when the subjects and objects co-occur at least 1-2k times even at early stages of pretraining. The results also indicate that the frequency threshold is related to the model size, and larger models tend to require smaller thresholds. Using the metrics that evaluate the quality of linear representations, the authors can predict the approximate frequencies of individual terms as well as the co-occurrence of terms in the pretraining corpus better than using the LMs' uncertainty alone.

**Strengths:**

This paper draws an interesting connection between pretraining term frequency to the formation of linear representation of factual recall relations. The fact that the formation of linear representations could happen at any pretraining stage is particularly intriguing. The experiments and results are easy to understand and the discussion of related work is comprehensive.

**Weaknesses:**

- Being able to predict the frequencies of individual terms as well as the co-occurrence seems to be a direct implication of high correlation and therefore does not sound like a major standalone contribution. Also, "Importantly, this regression model generalizes beyond the specific LM it was trained on without additional supervision.": the prediction results seem to be very noisy and not much better than the mean frequency baseline.
- Some claims are not properly justified:
  - Line 75: "This frequency threshold decreases with model size": This is only tested two model sizes (OLMo-7B and OLMo-7B), and the fact that GPT-J (6B) has as smaller threshold than OLMo-7B is an counterexample for this. Would be good just to be consistent with the rest of discussion to claim there is a connection to scale.
  - Line 93-94: "Linear representations form at predictable frequency thresholds during training, regardless of when this frequency threshold is met for the nouns in the relation." The term "predictable" can be understood as there is a strong correlation between the linear representation quality and the co-occurrence frequency. However, the fact that this threshold is predictable regardless of when this frequency threshold is met is not well supported by results. It is necessary to show that the threshold (mean causality >.9) is consistent across different checkpoints.
    - Line 319-320: "Regardless of pretraining step, models that surpass this threshold have very high causality scores." It would be nice if you could arrange results into different scatter plots for each pretraining stage and compare them, say by fitting a linear model and comparing their slopes and biases.
  - Line 100: The efficiency of the proposed searching tool is not well discussed.
  - Line 455: "Some relations, like star-constellation perform very poorly, possibly due to low frequency" Why low frequency is the cause?
  - Line 471: "Second, evaluating on the LRE features of a heldout model (scaled by the ratio of total tokens trained between the two models) maintains around the same accuracy," How do the results support "around the same accuracy"? If it is comparing Train OLMo and Train GPT-J in Table 1, the drop of accuracy is larger than the performance gap between LRE features and mean baseline. I am not sure if this entails "around the same accuracy".
  - Line 483: "In general, the regression transfers well, without performance deteriorating much (about 5%), suggesting LREs are encoding information in a consistent way across models." What results support this? Table 2 only shows a few examples which is insufficient for supporting the claim. Are there aggregated numbers of all pairs? How many of them have errors less than 5%?
- Some important details are missing from the experiments
  - The results in Figure 3 and Table 1 do not match: 1) why is the mean freq. baseline performance different? 2) why do LRE features (Table 1: 0.76) seem to perform better than LRE + LM (Figure 3: ~0.67) for OLMo, if Figure 3 shows the results for OLMo. The author should explain how are the numbers related to each other.
  - Table 2: 1) "Predictions are better for relations that are closer to those found in fitting the relation (country-related relations)" What does closer mean here? How did you measure this? 2) "Some relations, like star-constellation perform very poorly, possibly due to low frequency"
- Some figures and tables need to be more carefully explained:
  - The two left plots in Figure 2 need more explanations or presented in a better way. Specifically, why are some points darker than the others? What do the lines (light grey and dark grey lines) mean? Also, why do all dots for GPT-J have the same shape while the dots for the 2 left plots do not?
  - Table 1: "Train OLMo" and "Train GPT-J" are hardly self-explanatory, the authors should consider better ways to explain the settings.

**Questions:**

1. The experiments follow previous work and only analyze 25 relations. What are the reasons for not including other relations?
2. Section 4.3 is interesting to some degree but I am not sure about the implication of the results. Looks like it is just a description of what is observed. What is the research question you want to answer here?
3. Typos:
   1. Line 455 the regression model can "be" sensitive to ...
   2. Line 416: the paragraph does not end properly.

---

> ### Author Response · Authors · 2024-11-22
> **Thank you for the review, please see clarifying comments**
>
> Thank you for the in depth review. We’re glad the reviewer found the work interesting and easy to follow.The reviewer raises some good points, but we’d also like to clear some points up that seem to be misconceptions about the work. We’ve begun making some changes to make a few things more clear.
>
>
> >> Being able to predict the frequencies of individual terms as well as the co-occurrence seems to be a direct implication of high correlation and therefore does not sound like a major standalone contribution
>
> This is not necessarily true. Correlation tells us “how related are two variables?", while regression allows us to answer "How can we predict an output variable based on input variables?". Also, note that we capture more complex non-linear relationships with a more complex regression model (like the random forest used here - lines 361-362). Therefore, we can relate multiple input variables, like task hardness (measured by accuracy and log probs) in addition to LRE features.  In addition, even this simple correlation relationship has not been previously shown in previous work (Hernandez et al., 2024; Jiang et al., 2024; see related work on linear representations)  because it is typically difficult to count individual tokens in a pretraining corpus (especially throughout training). We are able to demonstrate as a proof of concept that the representations themselves reflect frequencies in the pretraining corpus, thus laying out future work on determining pretraining data exposure with more sophisticated measures.
>
> >> Line 75: "This frequency threshold decreases with model size": This is only tested two model sizes (OLMo-7B and OLMo-[1]B), and the fact that GPT-J (6B) has as smaller threshold than OLMo-7B is an counterexample for this. Would be good just to be consistent with the rest of discussion to claim there is a connection to scale.
>
> This is a fair point, and we have softened the claim about where this threshold lies across models. However, across  relations within model checkpoints we have strong evidence for some threshold existing, even if we can’t derive a model agnostic, scale derived threshold. We will highlight that it’s inconclusive where this threshold lies on any given model, but we report a few data points that point to some trend existing. At the same time, we can confidently say that frequency strongly correlates with linear representations forming.

---

> > ### Author Response · Authors · 2024-11-22
> > **Continued**
> >
> > > "Importantly, this regression model generalizes beyond the specific LM it was trained on without additional supervision.": the prediction results seem to be very noisy and not much better than the mean frequency baseline.
> >
> > We’d like to point out that we get negative results for subj-obj co-occurrence prediction which is a harder task. We revise this in section 5.3, but it’s likely that this problem requires more contextual information to get any reasonable performance. Another reason is that the data is highly concentrated around the mean, with a few outliers, which is demonstrated by the high accuracy with predicting the mean in this case.
> >
> >
> > > , the fact that this threshold is predictable regardless of when this frequency threshold is met is not well supported by results. It is necessary to show that the threshold (mean causality >.9) is consistent across different checkpoints.
> >
> > There seems to be a miscommunication around these results, because that is what our findings demonstrate. Consider the red triangle in the top right corner of the OLMo 7B graph in Figure 2: this is the country-largest-city relation for the checkpoint at only 10k steps. It is highly frequent and has a perfect causality score of 1.0. Considering only the red dots (the 10k step checkpoints), the correlation is clearly still visible, however with regards to the reviewer’s other point:
> >
> > >  It would be nice if you could arrange results into different scatter plots for each pretraining stage and compare them, say by fitting a linear model and comparing their slopes and biases.
> > This is good feedback, and we will add this to make our point much clearer. The reviewer raised a few points of concern with regards to this figure, and we believe this addresses them.
> >
> >
> > > The results in Figure 3 and Table 1 do not match:
> > > The author should explain how are the numbers related to each other.
> > > Table 1: "Train OLMo" and "Train GPT-J" are hardly self-explanatory, the authors should consider better ways to explain the settings.
> >
> > There are differences in these settings:‘Train OLMo’ refers to the setting where we fit the regression on OLMo data and evaluate it on GPT-J data. In this setting we are explicitly testing robustness to the difficult setting of heldout relations and a heldout model. In Figure 3 we are showing the setting of fitting and evaluating on heldout relations on the same model. Note that we must filter the datasets so that there is no overlap between seen examples from relations, and must only consider data that appears in both the PILE and Dolma, thus making the baselines that we are comparing completely different. We can discuss the differences more in the paper/appendix. Please let us know if this is still unclear, but essentially, these baselines aren’t meant to be compared.
> >
> >
> > >> Table 2: 1) "Predictions are better for relations that are closer to those found in fitting the relation (country-related relations)" What does closer mean here? How did you measure this?
> >
> > Here “closer” is a qualitative term describing the output domain of the relations trained vs. tested on. For example, fitting the regression on relations that output people’s names vs. constellation names. This is a fair point to raise, we have updated the paper to better explain these terms.
> >
> > >> Are there aggregated numbers of all pairs? How many of them have errors less than 5%?
> >
> > This is referring to a drop in 5% accuracy from OLMo (70%) to GPT-J (65%), i.e., performance is maintained across this model pair. We’ve updated this to say 5% within-magnitude accuracy.
> >
> > To summarize the revisions based on these comments: We improve the explanations for the differences in settings for the regression results and provide clearer breakdowns of the datasets used.
> >
> >
> > >> The experiments follow previous work and only analyze 25 relations. What are the reasons for not including other relations?
> >
> > In response to this and other reviewer comments, we will include an analysis on commonsense relations as well, which may provide additional insights on how/when linear representations form. We currently have the counts for these, and will report the results on these relations in the upcoming days
> >
> > >> Section 4.3 is interesting to some degree but I am not sure about the implication of the results. Looks like it is just a description of what is observed. What is the research question you want to answer here?
> >
> > The specific research question is “How does accuracy relate to the presence/absence of linear representations of relations?”. We were surprised that this trend had not been reported anywhere before, and found the relationship to be quite strong, however it is still unclear whether a linear representation causes or is necessary for high performance. The answer to this question has important implications for measuring specific model capabilities, so we wanted to highlight what we found as a starting point for future work.

---

> > > ### Author Response · Authors · 2024-11-22
> > > **Details on Efficiency**
> > >
> > > > Line 100: The efficiency of the proposed searching tool is not well discussed.
> > >
> > > We can expand on this. Our approach is 10-100x faster than reference approaches we tried (sliding window, naive search with np.where). Consider a matrix of size 4096x4096 which represents a batch of 4096 sequences of length 4096. We need to search that batch for any occurrences of any of the 10k entities (subjects and objects) in our dataset, which may be represented by multiple tokens each. A big part of the speedup comes from not having to store intermediate variables as python objects and from searching for many entities in parallel. Because the code runs entirely in C++, we can drop the GIL (Global Interpreter Lock) and parallelize multiple threads instead of processes (which take longer to instantiate, and need their own memory). At the end of the call, a numpy array with the exact indices across thousands of sequences in which the tokens. This would be impossible to do in reasonable time (years) in Python alone, but using Cython, our implementation can be called directly as a Python module in a standard environment, and is totally agnostic to the specific data passed into it. Please let us know what specifically the reviewer considers to be a proper discussion of the efficiency of our approach.

---

> > > > ### Comment · Reviewer_juu2 · 2024-11-26
> > > >
> > > > It is possible to compare the efficiency between the proposed tool to WIMBD? Actually this is my original question. There is nowhere in the paper mentioned sliding window or naive search with np.where, so i wasn't sure about what you mean efficient.

---

> > > > > ### Author Response · Authors · 2024-11-26
> > > > >
> > > > > We are glad the reviewer appreciates the changes we've made so far. Thank you for the continued feedback. We will address the question on the difference between the correlation and regression results, whether we should make a stronger claim on the connection to accuracy, and then move on to the remaining cosmetic questions.
> > > > >
> > > > > > Thanks for the explanation. I agree that "even this simple correlation relationship has not been previously shown in previous work", and I think this main finding of the paper is very interesting, as I said in the strengths. The point I am trying to understand here is, high-correlation seems to imply that you can "predict an output variable based on input variables". Consider a case where two variables are nearly perfectly correlated, isn't it obvious that one can fit a linear model that takes one as input and another as output?
> > > > >
> > > > > As we understand it, the reviewer is asking whether the results in Figure 2, that linear representations and pretraining frequency correlate very strongly entails the results that we can predict term frequency from LRE measurements. A very important distinction between the two sections is that we move from testing whether average frequency for terms in a relation correlate with an LRE being effective for that relation, to testing whether we can use the LRE's effectiveness on a given datapoint *to predict the frequency of that individual term*. Evidence that these two things are not the same is also present in the paper: subject-object frequency correlates more strongly with the LRE appearing than object frequency (.82 vs. .59 respectively), but predicting object frequency for an individual datapoint is much more effective than trying to predict the subject-object co-occurrence frequency compared to the mean baseline. Even disregarding this fact, the reviewer is entitled to think this isn't an interesting use case, but we think these experiments are necessary to show that there is practical significance to the correlational findings.
> > > > >
> > > > > > incontext learning accuracy on certain tasks has a strong correlation to the pretraining term frequencies. And in this work you show that "the presence/absence of linear representations" is strongly correlated to frequencies as well. Then isn't it already implied that there should be some correlation between the accuracy and "the presence/absence of linear representations"? And the an obvious hypothesis for "however it is still unclear whether a linear representation causes or is necessary for high performance" is the pre-training term frequency is the common cause.
> > > > >
> > > > > We were hoping to see a clear relationship where LREs form right before/after accuracy jumps, but we couldn't make a strong case for this. Are we overcomplicating this point? Yes, pretraining frequency seems to be the common cause, but we are wondering if the model is only accurate **because** of the presence of the linear structure (i.e., it won't be accurate unless these form). We are definitely receptive to feedback on this point.
> > > > >
> > > > > >it would be helpful if the authors can be more specific about the changes made, say by highlighting the changes.
> > > > >
> > > > > Please see the general comment
> > > > >
> > > > > We will now address the cosmetic questions:
> > > > >
> > > > > > why are some points darker than the others? What do the lines (light grey and dark grey lines) mean?
> > > > >
> > > > > This is now mentioned directly in the caption of figure 2, so the reader no longer has to go back and forth:
> > > > >
> > > > > "Symbols represent different relations. Highlighted relations are shown in darker lines." As already mentioned in the footnote, these are  ‘country largest city’, ‘country currency’, ‘company hq’, ‘company CEO’, and ‘star constellation name’. We chose these because they occupy different ranges of frequencies, to highlight the relationship.
> > > > >
> > > > > > Also, can you modify the paper to reflect our discussion: elaborate on the setting of ... Figure 2 (see my previous response)?
> > > > >
> > > > > We are currently migrating data between computer clusters and we do not have time to recreate the graphs in Figure 2 before the discussion period ends. Thank you for understanding, but we really like this cosmetic change and will definitely make it happen for the final version! Still we believe all of the data
> > > > >
> > > > > > why do all dots for GPT-J have the same shape while the dots for the 2 left plots do not?
> > > > >
> > > > > Each shape represents a relation. This is to visually help the reader look at the progress across checkpoints for a given relation on average. GPT-J does not have checkpoints so the same shape was used, but in hindsight, we agree we should show these relations! Again, we are migrating data, apologies that we can't make this change immediately.
> > > > >
> > > > > > elaborate on the setting of Table 1 (what does Train OLMo Train GPT-J mean)
> > > > >
> > > > > We have updated the table to say "Eval on X" instead of "Train on Y" to be a little more descriptive. In the caption of table 1, we added "Eval. on GPT-J means the regression is fit on OLMo and evaluated on GPT-J."

---

> > > > > > ### Author Response · Authors · 2024-11-26
> > > > > > **Comparison to WIMBD efficiency**
> > > > > >
> > > > > > > It is possible to compare the efficiency between the proposed tool to WIMBD?
> > > > > >
> > > > > > At the current point in time, we can not compare these efficiencies directly, but we can look into it. However, consider that WIMBD requires indexing an entire corpus before searching it. Our method searches individual batches for token occurrences and is better suited to track counts across training time. WIMBD is effective for repeated searches across the same corpus, while our method is flexible to work with new data without requiring reindexing (such as adding training data to a mix), so they serve different purposes.

---

> > > > > > > ### Comment · Reviewer_juu2 · 2024-12-02
> > > > > > >
> > > > > > > I appreciate the authors' response. Many of my concerns have been addressed. Although I still think many of the discussions and analyses are a bit repetitive (in-context learning accuracy, and predicting how often a term was seen in pertaining), the paper indeed provides interesting insights. Therefore I have increased my score accordingly.

---

> > > ### Comment · Reviewer_juu2 · 2024-11-26
> > >
> > > > There seems to be a miscommunication around these results, because that is what our findings demonstrate. Consider the red triangle in the top right corner of the OLMo 7B graph in Figure 2: this is the country-largest-city relation for the checkpoint at only 10k steps. It is highly frequent and has a perfect causality score of 1.0. Considering only the red dots (the 10k step checkpoints), the correlation is clearly still visible, however with regards to the reviewer’s other point: It would be nice if you could arrange results into different scatter plots for each pretraining stage and compare them, say by fitting a linear model and comparing their slopes and biases. This is good feedback, and we will add this to make our point much clearer. The reviewer raised a few points of concern with regards to this figure, and we believe this addresses them.
> > >
> > > Can you add the figure to the paper or appendix? Currently, Figure 2 and its caption are hard to understand. Specifically: "why are some points darker than the others? What do the lines (light grey and dark grey lines) mean? Also, why do all dots for GPT-J have the same shape while the dots for the 2 left plots do not? What do different shapes mean (should be either explained in the caption or represented in the legend)?"
> > >
> > > > This is referring to a drop in 5% accuracy from OLMo (70%) to GPT-J (65%), i.e., performance is maintained across this model pair. We’ve updated this to say 5% within-magnitude accuracy.
> > >
> > > Thanks for clarifying, but 70% (Figure 1) and 65% (Table 1) are not mentioned anywhere this paragraph and the mention of Table 2 surrounding makes it even more misleading. Consider referring to the numbers more directly.
> > >
> > > > The specific research question is “How does accuracy relate to the presence/absence of linear representations of relations?”. We were surprised that this trend had not been reported anywhere before, and found the relationship to be quite strong, however it is still unclear whether a linear representation causes or is necessary for high performance. The answer to this question has important implications for measuring specific model capabilities, so we wanted to highlight what we found as a starting point for future work.
> > >
> > > The reason I am confused about the section is that existing work ([1] and the ones you cited) have already shown that the incontext learning accuracy on certain tasks has a strong correlation to the pretraining term frequencies. And in this work you show that "the presence/absence of linear representations" is strongly correlated to frequencies as well. Then isn't it already implied that there should be some correlation between the accuracy and "the presence/absence of linear representations"? And the an obvious hypothesis for "however it is still unclear whether a linear representation causes or is necessary for high performance" is the pre-training term frequency is the common cause.
> > >
> > > [1] Razeghi Y, Logan IV R L, Gardner M, et al. Impact of pretraining term frequencies on few-shot reasoning[J]. arXiv preprint arXiv:2202.07206, 2022.

---

> > > ### Comment · Reviewer_juu2 · 2024-11-26
> > > **Be more specific on the changes made to the paper**
> > >
> > > I appreciate the effort the authors made to revise the submission based on the feedback. But it would be helpful if the authors can be more specific about the changes made, say by highlighting the changes.
> > >
> > > Also, can you modify the paper to reflect our discussion: elaborate on the setting of Table 1 (what does Train OLMo Train GPT-J mean) and Figure 2 (see my previous response)?

---

> > ### Comment · Reviewer_juu2 · 2024-11-26
> >
> > > Being able to predict the frequencies of individual terms as well as the co-occurrence seems to be a direct implication of high correlation and therefore does not sound like a major standalone contribution
> >
> > Thanks for the explanation. I agree that "even this simple correlation relationship has not been previously shown in previous work", and I think this main finding of the paper is very interesting, as I said in the strengths. The point I am trying to understand here is, high-correlation seems to imply that you can "predict an output variable based on input variables". Consider a case where two variables are nearly perfectly correlated, isn't it obvious that one can fit a linear model that takes one as input and another as output? I understand the other way around is not necessarily true: being able to fit a complicated regression model doesn't imply that two variables are highly correlated, e.g. y=sin(x). Correct me if I misunderstood some of your arguments.

---

### Official Review · Reviewer_zXUa · 2024-11-02

**Soundness:** 3
**Presentation:** 3
**Contribution:** 3
**Rating:** 6
**Confidence:** 4

**Summary:**

This research paper explores how the frequency of certain words appearing together in the data used to train a language model (LM) affects the LM's ability to learn simple, linear rules for representing facts. The authors found that the more often two words related to a fact appear together in the training data, the more likely the LM is to learn a simple rule to represent that fact. This discovery helps to understand how LMs learn factual information and could be used to figure out what kind of data was used to train secret LMs. The authors also created a tool to help others count how often words appear together in large datasets

**Strengths:**

The study finds a strong correlation  between the average co-occurrence frequency of subjects and objects within a relation and the quality of linear representations (LREs) formed for that relation. This correlation surpasses the individual correlations with subject frequencies or object frequencies, highlighting the significance of subject-object co-occurrence.

The study focuses uses Linear Relational Embeddings (LREs), which effectively approximate the computations performed by an LLM to predict objects in factual subject-relation-object triplets. This paper builds upon this research by examining how the frequency of subject-object co-occurrences in pretraining data directly impacts the emergence and quality of these LREs

The paper introduces a promising technique for analyzing the pretraining data of closed-source models by leveraging the connection between linearity and frequency.

**Weaknesses:**

The paper presents valuable findings, however they should provide some discussion along the following directions:

(a) The paper primarily focuses on Linear Relational Embeddings (LREs) as a representative class of linear representations in LLMs. However, LLMs might employ various other forms of linear or non-linear structures to encode information. This focus on LREs could limit the generalizability of the findings to other types of representations. Is there any strong hypothesis to strict to LREs?

(b) While the study demonstrates that LRE features can be used to predict the frequencies of individual terms with reasonable accuracy, predicting the frequency of subject-object co-occurrences is challenging. The regression models achieve only marginal improvements over baseline performance in this task.  Integrating additional features might be helpful here.

(c) The study analyzes a set of 25 factual relations from the Relations dataset. However, LLMs are trained on vast and diverse data, encompassing a much wider range of relations and concepts. Expanding the scope of analysis to encompass a broader range of relations would provide a more comprehensive understanding of the role of frequency in shaping LLM representations.

(d)  The paper focuses primarily on the frequency of terms in the pretraining data. However, other factors, such as the context in which terms appear, the syntactic structure of sentences, or the semantic relationships between words, could also influence the formation of linear representations. For example, LLMs are proven to not do well is facts  are stored in templates , as it tend to remember the template and not the facts. The proposed approach may not be applicable in those scenarios.

**Questions:**

Refer the previous section.

It would be good if authors can dedicate a section to discuss the potential impact of confounding factors, such as context, syntax, and semantics. Explain why controlling for these factors is challenging in the current study but emphasize the importance of future work to disentangle their effects from the influence of frequency.

---

> ### Author Response · Authors · 2024-11-22
> **Thank you for the review and suggested discussion points**
>
> Thank you for the detailed review and we’re glad the reviewer views our data analysis tool as a promising way to inspect closed-data LMs and that the findings are valuable.  We are happy to elaborate on the points brought up here:
>
> a.) The reviewer brings up the generalizability of these findings to other representational forms (linear, affine, non-linear). LREs have a particularly nice property in that they can capture relationships encoded as affine, linear, or translation transformations. We don’t explore constraining LREs in any particular way here (see, e.g., the translation baseline in the Hernandez et al. paper to get an idea which ones work with only a bias term). We agree more discussion about non-linear features will be useful, and have added some discussion about Csordas et al., 2024 (https://arxiv.org/abs/2408.10920v1) in lines 513--514, which finds evidence against the strong version of the linear representation hypothesis. They find an example of non-linear representations in a recurrent network. Our specific question is about how/when linear representations form, and want to be clear that we don’t rule out the existence of more complex features.
>
> b.) The reviewer brings up the relatively lower accuracies for predicting subject-object co-occurrences from representations compared to predicting object only frequencies. We indeed find positive results for predicting object occurrences and negative results for predicting subject-object occurrences from LRE features. Besides the fact that it is a much more difficult problem than predicting object occurrences alone, we can add more discussion on why we think this is. One reason this might be the case is that the distribution of subj-obj occurrences is very tight, with a few large outliers. For example, with language-of(Russia)=Russian, Russia and Russian co-occur 1.2M times, far outside the mean of around 100k for this relation. This makes it so the model can get good accuracy (near 70%) without capturing outliers. In terms of additional features that could be helpful: the pointwise mutual information between subject and object from a reference dataset may improve performance.
>
> c.) In response to using other relation types: Although we constrain our analysis to factual relations, these capture quite a large range of topics that would be memorized by models (geography, companies, familial relationships, occupations, media knowledge, etc.). However, to broaden our approach, we are adding analysis of commonsense relations as well (task-done-by-tool, fruit-inside-color, etc.). We think these might point to an interesting reporting bias, as some of these relations would be predicted as having low subj-obj co-occurrences (Paik et al., 2021 https://aclanthology.org/2021.emnlp-main.63/) . We appreciate this suggestion and will add these results in the coming days
>
> d.) see below:
>
> >>”other factors, such as the context in which terms appear, the syntactic structure of sentences, or the semantic relationships between words, could also influence the formation of linear representations”
>
>  We agree with this point and have added discussion in the Limitations section. In short, given the consistency of the ‘linearization’ across relations, we would predict that this has minimal impact. Still, we will leave room to describe possible variables we don’t account for.

---

### Official Review · Reviewer_J8rP · 2024-11-04

**Soundness:** 3
**Presentation:** 3
**Contribution:** 3
**Rating:** 6
**Confidence:** 3

**Summary:**

This paper explores the question of why linear structures form in LLMs by investigating the connection between training data frequency and the formation of linear representation, focusing specifically on factual recall relations. The study reveals that (1) the formation of linear representations is strongly correlated with subject-object co-occurrence frequency, and (2) the presence of linear representations can help predict relation frequency. Experiments are conducted using OLMo-1B, oLMo-7B, and GPT-J to validate these findings.

**Strengths:**

- Exploring the origin of linear representation is an important question in LM interpretability. This work identifies a correlation between linear representations of factual recall relations and the subj-obj co-occurrence frequency in pretraining.
- This paper investigates the relationship between few-shot accuracy and the existence of a linear representation.
- Using the existence of linear representations to predict the frequency of terms in the pretraining corpus is interesting.

**Weaknesses:**

- The scope of the work is somewhat limited, as only 25 factual relations are investigated. It is unclear whether the identified correlation is also valid for other relation types. Expanding the analysis to include more factual relations and other types of relations could further enhance the robustness of the findings and potentially offer additional insights.
- The linear representation seems to be affected by the context in LREs (e.g., four "X plays the Y" examples before the fifth one. Are the findings universally applicable to LLM generation without involving ICL formats?

**Questions:**

Please see weakness.

---

> ### Author Response · Authors · 2024-11-22
> **Thank you for the review. Update on relations tested**
>
> Thank you for your review and your questions, we’re glad the reviewer finds the work interesting and the results important.
>
> >>The scope of the work is somewhat limited, as only 25 factual relations are investigated. It is unclear whether the identified correlation is also valid for other relation types. Expanding the analysis to include more factual relations and other types of relations could further enhance the robustness of the findings and potentially offer additional insights.
>
> While we believe the 25 relations is not particularly limited (and covers a wide range of domains: geography, companies, familial relationships, occupations, media knowledge, etc.) we agree more relations will be helpful. We focus on factual relations because using subj-obj. Co-occurrences as a proxy for mentions is most strongly motivated by prior work (Elsahar et al., 2018), but with the relationship we have currently established, it will be interesting to expand no that.
>  We are expanding our analysis to the commonsense relations in the Relations dataset. These are relations like “task done by tool” (shovels are used to dig, e.g.). We have collected the counts for the 8 commonsense relations from this dataset, and will post the analysis in our rebuttal in the next few days.
>
>
> Whether this property holds outside of ICL templates, we know that they do in the zero-shot setting (if this could be considered non-ICL, the template is still technically the same). We have these results but did not include them in the paper because it’s shown in prior work (Hernandez et al., 2024)

---

> > ### Comment · Reviewer_J8rP · 2024-11-26
> >
> > Thank you for your responses, which have addressed my second concern. I would like to maintain my positive assessment of this work.

---

### Official Review · Reviewer_eSj4 · 2024-11-04

**Soundness:** 3
**Presentation:** 4
**Contribution:** 3
**Rating:** 6
**Confidence:** 3

**Summary:**

The authors investigate the correlation between linear representations and pre-training data frequency in language models. The work is conducted on recent findings that the linearity of different types of relations varies significantly depending on the specific relationship. Existing work does show that language model exhibit such linear structures, but do not reveal the underlying reason why some relations exhibit such structure while other do not. The main contribution of this work is to empirically draw the correlation between such linear structure and data frequency. It shows that that linear representations for factual recall relations are related to mention frequency and the model size. In addition, more detailed results show that linear representations form at predictable frequency thresholds during training, which allows the prediction of term frequencies in the training data. Finally, the authors release a tool for searching through tokenized text for understanding training data characteristics.

Overall, the findings are insightful for understanding linear representation structures in language models. This empirical study complements existing theoretical evidence on the same subject. It provides a perspective to the problem, which can be among many other factors in driving the formation of linear structures. On the utility side, the findings can be used for understanding training data, which are typically not published for current LLMs.

**Strengths:**

A perspective for understanding the reason that some features from LLMs demonstrate linear structures while others do not.

A tool for search through tokenized text to support the understanding of training data.

**Weaknesses:**

It provides one perspective to the problem empirically, with a specific set of metrics. While giving useful information, the depth of understanding and the utility domain is constrained mostly to the correlation between term frequency, the model size and the linear structure.

**Questions:**

Could there be some theoretical discussion on the training dynamics and the frequency thresholds?

One related work is

Guangsheng Bao, Zhiyang Teng, and Yue Zhang. 2023. Token-Level Fitting Issues of Seq2seq Models. In Proceedings of the 8th Workshop on Representation Learning for NLP (RepL4NLP) at ACL 2023. Toronto, Canada from July 9th to July 14th, 2023.

which also discusses the correlation between term frequencies and model accuracy.

---

> ### Author Response · Authors · 2024-11-22
> **Thank you for the review**
>
> Thank you for the review. We’re glad the reviewer found the work insightful. We hoped to fill a gap in understanding that theoretical work could not address, so we’re also happy this came across in the paper. To address your question, Ethayarajh et al,. https://aclanthology.org/P19-1315.pdf (and the cited related work within) point to frequency driving structure in static word embeddings, as well as the training objective driving linear representations in LLMs in Park et al., 2024 (https://arxiv.org/abs/2403.03867). We have provided more discussion synthesizing these ideas in our own work in the updated pdf in the discussion section. Thank you for pointing out some related work we missed, as well.
> While our analysis is mostly correlational, however, we show that the same trends hold for two model families (OLMo and GPT-J). If the reviewer has specific feedback on what metrics or experiments would broaden the impact of the paper, we would be happy to consider implementing them, if feasible.

---

> > ### Comment · Reviewer_eSj4 · 2024-11-25
> >
> > Thank you for the rebuttal and the additional details. I find that this is a solid and dedicated contribution to a specific issue, and the related discussion was useful, which increased my contribution score. I do not have a specific set of experiments in my mind for extending the broader impact, but maintain my overall borderline positive standing.

---

### Author Response · Authors · 2024-11-26
**Summary of changes made during the discussion (and pending cosmetic changes)**

Thank you to the reviewers for their work in evaluating this work. We are happy that the reviewers agreed the work was interesting and valuable, as well as important for model interpretability. Additionally, we are glad that reviewers were generally excited about the potential use of our findings in tools for making inferences about the training data of language models (LMs). We will outline the contributions, points that reviewers made, and our responses here:

## Contributions
1. We identify a correlation between linear representations (in the form of linear relational embeddings (LREs), see [Hernandez et al., 2024](https://arxiv.org/abs/2308.09124)) of factual recall relations and the subject-object co-occurrence frequency in pretraining.

2. We introduce a tool for quickly searching and counting token occurrences in training data batches that offers more flexibility than existing tools.

3. We leverage the connection between linear representational structure and frequency to show that we can use the presence and 'strength' of an LRE to predict the pretraining frequency of *individual terms* in the pretraining data, allowing us to make inferences about the pretraining data of open-weights models.


## Changes

The biggest change we made was **adding more relations**. There were concerns that 25 relations is not enough or that limiting to factual relations was not telling a general enough picture. We want to emphasize that this is not a small dataset, across the 25 relations, we had **over 10,000** unique subjects and objects. We also focus on factual relations following prior work ([Elsahar et al., 2018](https://aclanthology.org/L18-1544/)) which finds that using subject-object co-occurrences is a good proxy for counting factual mentions. Still, there are interesting questions around how the analysis extends to other relations. To answer this, we included 8 additional commonsense relations. These are: fruit_inside_color, fruit_outside_color, object_superclass, substance_phase, task_done_by_person, task_done_by_tool, word_sentiment, work_location. We added these as Appendix F. While we find interesting relationships between frequency and causality that mirror the factual relations, we also raise issues with using subject-object co-occurrences as counts for some of these relations.

We outline the remaining changes requested by reviewers below:

* We added more discussion around prior theoretical results to the Discussion section (reviewer eSj4)

* Included further discussion around non-linear features (reviewer J8rP)

* Qualified the claim about whether we could derive model-agnostic frequency thresholds for when linear representations form (multiple places). We agree we do not have enough data to support the strong version of this claim. (reviewer juu2)

* Updating Table 1 to be more descriptive of the setup where we test cross-model generalization of the regression model we fit (juu2). We updated the caption to this table as well to reflect this

* Updating the caption in Figure 2 to be more descriptive of which features we are discussing

* Clarifying which numbers we are comparing when discussing generalization performance of the regression (L482-483)

We received advice for making Figure 2 more readable from reviewer juu2. We believe this is very helpful feedback and will *definitely* make the changes in the final revision, but we won't be able to make these changes before the rebuttal deadline. Note however, that this is purely presentational, and the relevant data is presented in the current draft.

Once again thank you for a very productive round of reviews

---

### Meta-Review · Area_Chair_X45U · 2024-12-21

**Metareview:**

his paper explores the Linear Representation Hypothesis in LLMs. Specifically, it aims to demonstrate a correlation between the cooccurrences of a subject and object and their relationship being represented in a linear way.  The paper establishes thresholds in different models beyond which this happens. It then looks to go the other way and predict pre-training frequencies given representations. Adding LRE features improves performance here.

The core question of this paper is very timely and interesting. The paper is clearly written and presents what I would characterize as medium-strong evidence of the correlation between frequency and linearity. The types of representations formed by LLMs and what factors of training cause them to emerge is a very relevant question. This paper advances the state-of-the-art in our understanding of these points.

The biggest issue with this paper is the scope and impact of the results. juu2 brings up the transfer to another LLM; the results in Table 1 aren't all that strong.  The restriction to a particular set of relations (and to KG relations in general) is also somewhat limiting. I don't think there's one right answer for how much the paper needs to engage with on this front, but making stronger and more general claims would of course make it stronger. As a more minor point, juu2 points out that connections of the paper's hypotheses with model size can't necessarily be drawn from the given data.

Taken together, all of this contributes to an impression that this paper has some important and useful results, but it might not be the last word on this topic. A scaled-up set of experiments, scaled out to different settings, may find something new and different here.

Finally, the paper explores the same question in a few different ways. For instance, the ability to predict the frequency is mostly a consequence of correlation (juu2), which lessens its impact a bit.

**Additional Comments On Reviewer Discussion:**

J8rP points out limitations of the task scope and task formatting, which are somewhat addressed in the response and new experiments.

juu2 brings up a number of points about the presentation and interpretation of the results, including the points mentioned above about correlation. Most of these presentational concerns are addressed.

---

### Decision · Program_Chairs · 2025-01-22

Accept (Poster)